# Effects of high-intensity interval training on strength, speed, and endurance performance among racket sports players: A systematic review

**Yixuan Liu**, **Borhannudin Bin Abdullah** *, **Hazizi Bin Abu Saad**

Faculty of Educational Studies, Department of Sports Studies, Universiti Putra Malaysia, Serdang, Malaysia

* borhannudin@upm.edu.my

**Data Availability Statement:** All relevant data are within the paper and its Supporting information files.

## Abstract

This study aims to present a critical review of the existing literature on the effects of High-Intensity Interval Training (HIIT) on strength, speed, and endurance performance among racket sports athletes. This study conducted a systematic literature review by PRISMA guidelines. Various well-known academic and scientific databases were used for research collection, including PubMed, EBSCOhost, Scopus, Web of Science, and Google Scholar. Out of 27 relevant studies, 10 were selected for inclusion in this systematic review, all meeting the required inclusion criteria. The quality of each study was assessed using the PEDro scale, with scores ranging from 3 to 5 for the selected studies. HIIT was found to improve racket players' VO2 max (maximum oxygen uptake), running and repetitive sprint performance, jumping performance, and hitting speed during play. Current findings indicate that HIIT can significantly benefit athletic performance. Long-term HIIT allows athletes to enhance their power while improving crucial variables related to both aerobic and anaerobic endurance. This anaerobic endurance and explosive power type is particularly vital for racket sports players. For example, athletes in table tennis and badminton must exert maximum effort during high-intensity middle and back-court play. Racket athletes also need to maintain a stable state while preserving ball speed and positioning, and must quickly recover to prepare for the next rally. This training mechanism can assist athletes in honing their skills and achieving more efficient hitting quality. Therefore, this paper recommends that racket sports athletes incorporate HIIT into their regular training routines. The suggested frequency is three times per week, with each training session lasting 30–40 minutes, and a total duration of six to eight weeks. **Trial registration. Systematic Review Registration:** [https://inplasy.com/], identifier[INPLASY20230080].

## 1. Introduction

Racket sports are a subset of ball games and may also be classified as those games where tools, such as rackets, are used to propel or strike a ball [1,2]. These matches typically involve two to four players and are centered around the objective of striking the ball so that the opponent

**Funding:** The author(s) received no specific funding for this work.

**Competing interests:** The authors have declared that no competing interests exist.

cannot return it [3,4]. Racket sports fall into two main categories: net sports, played on designated courts like tennis, badminton, and table tennis, and non-net sports, played on shared courts, such as squash [2,5].

Racket sports necessitate a combination of high-intensity interval exercise and low-intensity exercise. To succeed, athletes in this field must possess a blend of speed, strength, and exceptional aerobic endurance [6,7]. During competitions, the body alternates between periods of high-intensity work, using intramuscular phosphate and glycolysis to replenish energy stores and restore homeostasis [8,9]. The ATP-CP energy system is vital for brief bursts of energy and shifts as energy demands change during the match. Initially, during vigorous exercise, the body relies on the ATP-CP system, emphasizing the importance of explosive power [3,10]. However, as the activity continues, the rapid depletion of ATP and CP stores due to their limited capacity prompts a transition to the lactate system when these reserves prove inadequate for sustained physical exertion [4,11]. In racquet sports, the body perpetually shifts between energy systems as the intensity and duration of play evolve. Short, explosive movements over limited distances are reliant on the ATP-CP system, while protracted rallies and enduring competitions engage the lactate system [12–14]. Comprehensive fitness programs for racket players typically encompass training for both anaerobic and aerobic energy systems to enhance performance [15,16]. Some studies have analyzed athletes' technical movements during competitions, showing that the thighs push against the ground to convert gravitational, elastic, and chemical energy into kinetic energy [17–19]. The energy transfer from the lower to the upper limbs is facilitated by the core area, involving a continuous muscle contraction, thereby making strength vital for performance [8,12,16].

Racket players require speed and explosiveness to generate striking power [20]. A more powerful drive enables faster, more accurate, and more challenging ball hits for the opponent [21]. Strength also positively affects movement balance and injury prevention, helping athletes maintain proper body positioning and meet the demands of prolonged competition [22]. Furthermore, quick movements and reflexes are crucial for racket sports. Excellent speed allows for more effective court coverage and better-timed strikes [8,23]. Thus, speed is important for both defending against and launching attacks [24,25]. Moreover, the accumulation of lactic acid during high-intensity exercise can affect internal stability and metabolic processes, leading to fatigue [5,26]. Therefore, endurance is another key aspect; athletes' muscles must be conditioned to withstand prolonged, repetitive, high-intensity activity. Training programs that take endurance into account are essential for maintaining high-performance levels during matches [4,27].

The scientific method of strength training serves as the optimal guideline for enhancing athletes' sports performance [28,29]. Racket sports are categorized as middle-level in terms of exercise intensity, necessitating attention to both the anaerobic and aerobic capacities of athletes [8,12,30]. One study demonstrated that racket-playing athletes experienced an increase in average blood lactate levels as match intensity escalated [25]. This suggests that players tend to rely on a relatively high proportion of aerobic energy during matches, due to the extended intervals between games in racket sports [22,31]. Enhancing endurance can mitigate the risk of fatigue, thereby preserving accuracy and decision-making abilities. It also facilitates quicker recovery between points and helps players rebound from challenging situations [32].

HIIT reigns as a widely embraced and potent exercise methodology, celebrated for its capacity to significantly enhance cardiovascular endurance and foster overall strength [33–35]. At its essence, HIIT thrives on the rhythmic alternation of brief, vigorous exercise intervals with equally succinct interludes of rest or low-intensity activity [36]. This harmonious interplay of intense training and strategic recovery cycles bestows upon HIIT the mantle of a stellar choice for those who seek a profoundly efficient and impactful workout [37–39].

Simultaneously, the fitness industry boasts a plethora of terminology to define training methods featuring the cyclic integration of high-intensity workouts and periods of rest or low-intensity intervals [40,41]. Terms such as 'high-intensity interval exercise' and 'high-intensity intermittent training' align with the core principles of HIIT [40,42,43]. Nevertheless, HIIT has emerged as the prevailing and universally embraced terminology in the fitness community [43,44]. This consensus streamlines communication and fosters better understanding among coaches, athletes, and fitness enthusiasts, rendering it the primary choice for describing this training method in most contexts [18,35]. Current research on HIIT confirms its efficacy in improving individual strength and promoting muscle hypertrophy (muscle growth) in athletes [4,45]. HIIT activates fast-twitch muscle fibers, which are responsible for generating high levels of strength and power. This activation promotes muscle adaptation and an increase in strength [13]. Additionally, HIIT stimulates the production of anabolic hormones like testosterone and growth hormone, which are instrumental in muscle protein synthesis and, consequently, in muscle growth and hypertrophy [19]. HIIT also induces metabolic stress and causes micro-tears in muscle fibers, triggering physiological responses that promote anabolic signaling and growth factors, leading to increased muscle strength and hypertrophy [8,12,46]. HIIT engages both aerobic and anaerobic energy systems, offering athletes the opportunity to enhance their speed and power in racket sports by targeting high-strength muscle fibers [8,47]. HIIT not only improves anaerobic capacity but also emulates the speed and duration of activities specific to various sports, thereby aiding athletes in developing energy systems and muscle adaptations tailored to their needs [21,44]. HIIT can also significantly improve an athlete's VO2 max, representing maximal oxygen utilization during exercise. Elevated VO2 max levels indicate better aerobic capacity and endurance, enabling sustained submaximal-intensity training over extended periods [28,48]. Compared to traditional steady-state cardio or prolonged strength training sessions, HIIT workouts are notably shorter. These high-intensity, short-duration exercises lead to metabolic adaptations like increased mitochondrial density and enhanced glucose utilization, both crucial for strength development [29].

While existing literature confirms the impact of HIIT on sprinting and cardiorespiratory performance in adolescents [49], studies across various sports have shown associations with HIIT in metrics like VO2 max, shuttle performance, strength, repetitive sprinting, and jumping [37,40], however, there remains a gap in the literature concerning systematic reviews specifically addressing the physiological impacts of HIIT on athletes engaged in racquet sports. Therefore, the primary objective of this study is to explore the influence of HIIT on the strength, speed, and endurance performance of racket sports athletes, employing a systematic literature review approach.

## 2. Materials and methods

### 2.1 Protocol and registration

The Preferred Reporting Items for Meta-Analysis (PRISMA) guidelines were followed in this literature review to systematically gather, select, and analyze data. The review was registered on the INPLASY website at [https://inplasy.com/], under the identifier INPLASY202320080 [50].

### 2.2 Search strategy

This study adhered to the PRISMA statement in both its design and execution. A comprehensive literature search was conducted across four reputable academic databases: PubMed, EBSCOhost, Scopus, and Web of Science. Google Scholar and Reference were also used as search engines. For each independent database, a carefully crafted search query was implemented,

focusing on the title and abstract. The search employed predefined keywords, utilizing the following query: ("High-Intensity Interval Training" OR "HIIT" OR "High-Intensity Intermittent Training") AND ("Strength" OR "Power" OR "Speed" OR "Endurance") AND ("Table Tennis Players" OR "Tennis Players" OR "Badminton Players" OR "Racket Players").

## 2.3 Eligibility criteria

The systematic searches followed the PICOS framework, which encompasses the following key concepts: 1. Population, 2. Intervention, 3. Comparison, 4. Outcome, and 5. Study design. As outlined in Table 1, the PICOS framework served as the inclusion criteria for publications. The following specific requirements were considered to determine a study's eligibility for inclusion:

1. The research population must include racket players, such as those in table tennis, tennis, and badminton, irrespective of gender or age.

2. The intervention in the research should specifically focus on HIIT. Distinct and explicit comparisons with alternative training methods must be discussed separately.

3. HIIT should be compared with other training programs.

4. The study must examine at least one aspect of the effect of HIIT on strength, endurance, or speed in racket athletes.

5. The study should include experimental articles, which may consist of two-group controlled trials (randomized or non-randomized) or single-group trials.

## 2.4 Study selection

After two independent authors selected articles that met the inclusion criteria, this review employed an EndNote citation management system to identify and remove duplicates. The titles and abstracts of the papers were assessed by Liu and Borhannudin to determine their suitability for inclusion in this study. In cases where the two authors disagreed on the selection of an article, a third author conducted a comprehensive analysis of the full article to make the final decision.

## 2.5 Data extraction and quality assessment

Upon completing the data retrieval phase, the study extracted key insights from eligible research articles. These insights included crucial details such as author names, publication years, and extensive population characteristics like participant numbers, types, age groups, and gender distribution. Furthermore, meticulous documentation of intervention characteristics, including the type, specific measures employed, and frequency, was conducted [51]. To rigorously assess trial quality, the study used the well-established PEDro scale, originally

**Table 1. PICOS Eligibility criteria.**

| PICOS | Detailed Information |
|---|---|
| Population | Racket players (table tennis, badminton, tennis) |
| Intervention | High-intensity interval training (HIIT) |
| Comparison | HIIT vs. Other training programs |
| Outcome | The effect of HIIT on strength, endurance, and speed among racket players |
| Study design | Two-group controlled trials (randomized/non-randomized)/single-group trial |

proposed by Herbert and Elkins [52]. This scale evaluates four essential methodological aspects: randomization, blinding procedures, comparisons between study groups, and the robustness of data analysis.

The PEDro scale, built on the foundation of the Delphi list initially developed by Verhagen et al. [51], consists of 11 items, each contributing to a comprehensive assessment of methodological integrity. Two independent raters, specially trained for this task, systematically assessed the quality of trials within the PEDro database. Any discrepancies or conflicts that arose during this evaluation process were carefully considered and resolved through the input of a third, impartial rater [52].

In summary, the PEDro scale, with scores ranging from 1 to 10, serves as a valuable tool for gauging the methodological quality of the studies included. Higher scores indicate superior methodological rigor. Articles that received scores between 8 and 10 were classified as exhibiting excellent methodological quality. Those with scores ranging from 5 to 7 were considered to demonstrate a high level of research quality. Articles that scored between 3 and 4 were characterized as having moderate research quality, while those with scores below 3 were deemed to lack sufficient methodological rigor and were consequently excluded from this comprehensive study [51].

## 3. Results

### 3.1 Study selection

A total of 22 articles were identified during the initial database search: seven from Web of Science, nine from PubMed, four from EBSCOhost, and two from Google Scholar. Duplicate articles were meticulously removed using EndNote software. Subsequently, a second round of screening eliminated two non-full-text articles and three non-journal articles. In the third screening phase, 17 full-text articles were assessed for eligibility. Seven articles were excluded because they did not align with the subject area of interest. Ten relevant publications satisfied the inclusion criteria and were selected for qualitative synthesis. The detailed process undertaken in this study is visually depicted in Fig 1.

### 3.2 Study quality assessment

As presented in Table 2, the assessment conducted via the PEDro scale revealed that the average score for the data included in this study falls within the range of 3–5. While the quality of the incorporated studies is generally commendable, it is noteworthy that all these studies encountered challenges related to hidden allocation, blinding of participants, and criteria about evaluators, therapists, and intent-to-treat analysis. From among the included studies, five were able to specify eligibility criteria, ensure comparability of baseline groups, conduct comprehensive between-group comparisons, provide precise point estimates, and account for variability. The intervention under scrutiny involves HIIT, which inherently carries associated risks of professional and sports-related injuries, thus achieving complete blinding of participants, evaluators, and therapists becomes a formidable challenge. This highlights the need for future research endeavors to prioritize higher-quality study designs and elevate the level of evidence in this domain.

### 3.3 Participant characteristics

Table 3 outlines the key characteristics of the ten studies that met the inclusion criteria.

1. Athlete Categorization: One study focused on table tennis players [53]. Four studies centered around badminton players [54–56,58]. Five studies examined tennis players [24,36,57,59,60].

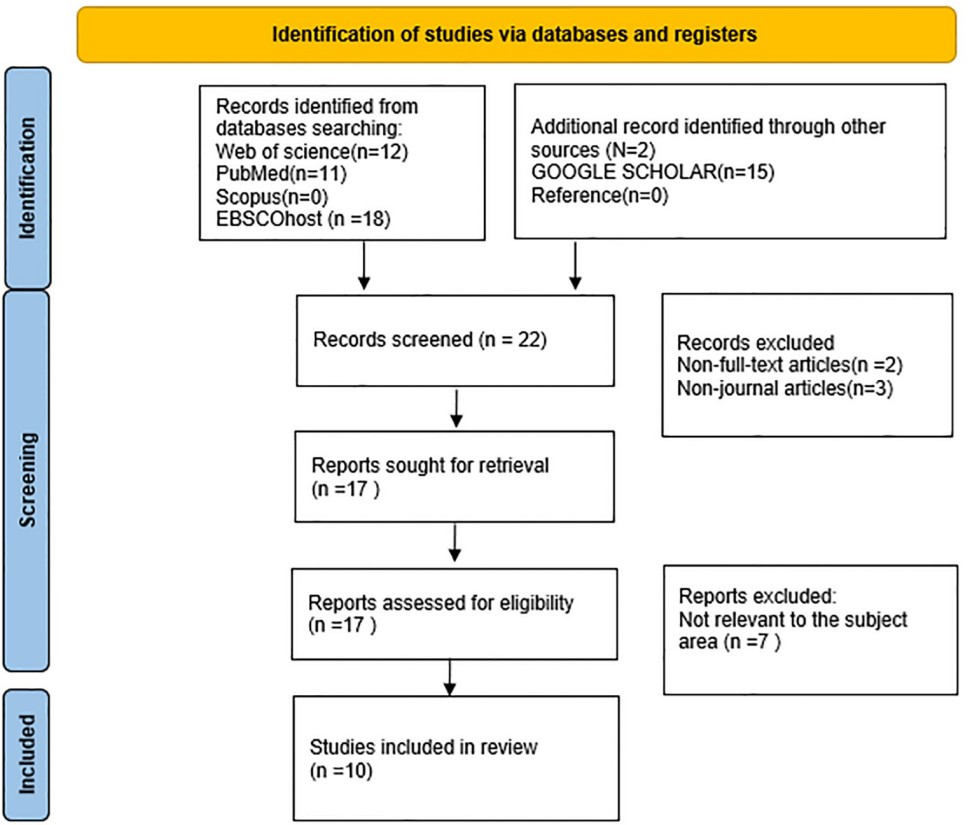

**Fig 1. The PRISMA flow chart for the search, screening, and selection strategy for the eligible studies.** *From*: Page MJ, McKenzie JE, Bossuyt PM, Boutron I, Hoffmann TC, Mulrow CD, et al. The PRISMA 2020 statement: an updated guideline for reporting systematic reviews. BMJ 2021;372:n71. doi: 10.1136/bmj.n71. For more information, visit: http://www.prisma-statement.org/.

2. Sample Size: Across these ten studies, a total of 227 participants were involved, with sample sizes ranging from 13 [59] to 32 subjects [55]. The median sample size was 20.5, and the mean was 22.3.

3. Gender: All ten studies focused on racket athletes [24,36,53–60]. Two studies specifically involved female athletes [53,57]. The remaining eight studies included males at three studies did not include data on subjects' weight, height, or BMI [36,57].

4. Age: All ten studies provided information on the age of their subjects, along with the age range [24,36,53–60].
   The analysis revealed that participants' ages ranged from 12 years [57] to 22 years [54].

5. Body Mass Index (BMI): Five studies reported both height and weight [54–56,58,60].

## 3.4 Intervention characteristics

Key characteristics of the intervention, including its type, duration, and frequency, were examined across the ten included studies. All studies employed HIIT as the primary intervention, with some studies referring to it simply as interval training.

**Table 2. Summary of methodological quality assessment scores.**

| Study | Eligibility Criteria | Random Allocation | Allocation Concealment | Baseline Comparability | Blind Therapist | Blind Assessor | Follow up | Intention to Treat Analysis | Between Group Comparisons | Point Measure and Variability | Total PEDro Score |
|---|---|---|---|---|---|---|---|---|---|---|---|
| TH,P (2017) [53] | 1 | 1 | 0 | 1 | 0 | 0 | 1 | 0 | 0 | 1 | 5 |
| Liu et al. (2021) [54] | 0 | 1 | 0 | 1 | 0 | 0 | 1 | 0 | 1 | 1 | 5 |
| Ko et al. (2021) [55] | 0 | 0 | 0 | 1 | 0 | 0 | 1 | 0 | 0 | 1 | 3 |
| Fuentes et al. (2021) [36] | 1 | 0 | 0 | 1 | 0 | 0 | 1 | 0 | 1 | 1 | 5 |
| Suppiah, 2019 [56] | 0 | 1 | 0 | 1 | 0 | 0 | 1 | 0 | 0 | 1 | 4 |
| Fernandez et al. (2012) [57] | 0 | 1 | 0 | 1 | 0 | 0 | 1 | 0 | 0 | 1 | 4 |
| Wee et al. (2017) [58] | 0 | 0 | 0 | 1 | 0 | 0 | 1 | 0 | 1 | 1 | 4 |
| Kilit and Arslan, (2019) [24] | 0 | 1 | 0 | 1 | 0 | 0 | 1 | 0 | 0 | 1 | 4 |
| Rodríguez et al. (2017) [59] | 1 | 1 | 0 | 1 | 0 | 0 | 1 | 0 | 0 | 1 | 5 |
| Fernande et al. (2017) [60] | 1 | 0 | 0 | 1 | 0 | 0 | 1 | 0 | 1 | 1 | 5 |

The intervention characteristics reported in the ten studies were based on the following:

1. Training Duration: The shortest intervention period was four weeks [36,55,57]. Minimum intervention expected 2 times [59]. Other studies had intervention durations of six weeks [24,57] and eight weeks [54,57], two study had an intervention period of 10 weeks [53,56].

2. Training Frequency: An analysis of the ten research reports revealed that training frequency ranged from two to three times a week. Specifically, one studie reported a training frequency of twice a week [58]. The remaining eight studies conducted training sessions three times per week.

3. Heart Rate: In 5 out of the 10 studies, heart rate was not explicitly documented, while two studies measured heart rate using the Hrmax method [54,55]. Another study utilized Team2 Pro as the criterion for heart rate measurement [53].

4. Interval Time: Nine of the ten studies employed a 30-second interval, whereas one study utilized a 15-second interval [57].

## 3.5 Outcome

In this study, the results were organized by categorizing racket athletes based on their performance in strength, speed, and endurance, all within the context of HIIT. Each author independently classified the papers according to the specific topics explored within the components of

**Table 3. Characteristics of the studies examined in the present review.**

| Study | Type of athletes | Population characteristics | Interventions | Type of exercise training | Measures index | Outcomes |
|---|---|---|---|---|---|---|
| TH, P. (2017) [53]. | adolescent table tennis players | EG1 = 10, CG = 10 Age:12±1.6yr., WT: 19.68± 2, HT; NR, CG = 12, | Freq.: 3 times/week, time: 2h-3h, length: 10 weeks Heart rate; Record the whole process Interval Time: 30s Heart Rate: NR Whether it meets the HIIT training intensity and interval time: Yes | High-intensity interval training group (EG), control group (CG) | Endurance; Heart rate (Team2 Pro), Filmate Pro (Vo2 max), Accoutered Plus meter(Blood lactate) | Vo2 max↑ Blood lactate↑ |
| Liu et al. (2021) [54] | Elite Badminton Player | Sex: F and M, EG1 = F/8M/8, CG = F/8M/8 Age: M/20.0 ± 1.3, F/20.5 ± 1.4, WT: M/73.8±6.9, F/62.6 ± 4.2, HT: M/179.6 ± 3.6cm, F/168.5 ± 4.2cm | Freq.: 3 times/week, time: NR, length: 8 Weeks Heart rate;50% to 90%HRmax Interval Time:30s Whether it meets the HIIT training intensity and interval time: Yes | High-intensity interval training (EG), traditional training (CG) | YO-YO IR2 intermittent recovery test, increasing load gas metabolism analysis, and lactate clearance rate test. | ventilatory anaerobic threshold↑, the ventilatory anaerobic threshold in the percentage of VO2max↑, lactate clearance↑, |
| Ko et al. (2021) [55] | Adolescent Badminton Players | Sex:M, EG1 = 16,CG = 16Age: M/20.0 ± 1.3, F/20.5 ± 1.4, WT: M/73.8±6.9, F/62.6 ± 4.2,HT: M/179.6 ± 3.6cm, F/168.5 ± 4.2cm | Freq.: 3 times/week, time: 30min, length: 4 weeks Heart rate; Maintain 50% HRmax to Beyond 90% HRmax Interval Time:30s Whether it meets the HIIT training intensity and interval time: Yes | High-intensity interval training (EG), Moderate continuous training (CG)) | Wingate test, isokinetic muscle function test | peak power↑ fatigue index↑ Heart Rate↔ Isokinetic Muscle strength↑ |
| Fuentes et al. (2021) [36] | Recreational Tennis Players | EG1 = 16, CG = 16 Age: 21.40 ±1.52,AEP = 0.8year WT:NR HT:NR | Freq.: 3 times/week, time: 0.78hours, length: 4weeks Interval Time:30s Heart rate: NR Whether it meets the HIIT training intensity and interval time: Yes | High-intensity interval training (EG), Stroop training (CG)) | Serve ball speed, Lower body muscular power, Spirometry, and Isometric hand strength, 、 | hitting speed↑ accuracy score, spirometry↔ |
| Suppiah et al., (2019) [56] | Collegiate athletes (badminton players) | Sex:M,Age:20±1,WT = 65.3 ±11, HT = 173.0 ± 5.3, EG = 9, CG = 9 | Freq.: 3 times/week, time: 30mins length: 10weeks Interval Time:30s Heart rate: NR Whether it meets the HIIT training intensity and interval time: Yes | High-intensity interval training (EG), traditional training (CG) | 20 m Multistage Fitness, Four Corner Agility, 20 m Sprint. | VO2max↑, sprinting ↑ |

*(Continued)*

**Table 3.** (Continued)

| Study | Type of athletes | Population characteristics | Interventions | Type of exercise training | Measures index | Outcomes |
|---|---|---|---|---|---|---|
| Fernandez et al. (2012) [57] | Tennis players | Sex: M, EG1 = 11, EG2 = 12, CG = 9, Age = 12.0 ± 3.6 years, WT = NR, HT = NR | Freq.: 3 times/week, time: 30min, length: 6weeks Interval Time:30s Heart rate: NR Whether it meets the HIIT training intensity and interval time: Yes | High-intensity interval training (EG1), repeated-sprint training (EG2), traditional training (CG) | incremental treadmill test, Hit and Turn Tennis Test, Vertical Jumping, Twenty-meter Sprint Run, Repeated-Sprint Ability Shuttle Test | VO2max↑, Hitting speed↑, 20-m sprint↔, 400m↑ |
| Wee et al. (2017) [58] | Collegiate athletes (badminton players) | Sex: M, EG = 9, CG = 9 Age: age = 20±1; WT = 65.3±11kg; HT = 173.0±5.3cm | Freq.: 3 times/week, time: NR length: 4weeks Interval Time:15s Heart rate: 165-185bpm Whether it meets the HIIT training intensity and interval time: Yes | High-intensity interval training (EG), traditional training (CG) | VO2 max Test, Wingate Ergometer Test, Countermovement Vertical Jump, Drop Jump, and Illinois Agility Test. | VO2max↑, mean power↑, reactive strength ↑ |
| Kilit and Arslan, (2019) [24] | Young Tennis Players. | Sex: M, EG = 14, CG = 15, Age: 13.8 ± 0.4 years, WT = NR, HT = NR | Freq.: 3 times/week, time: 20-30min length: 6weeks Interval Time:30s Heart rate: NR Whether it meets the HIIT training intensity and interval time: Yes | High-intensity interval training (EG), court tennis training (CG) | Maximum oxygen consumption, sprinting, jumping, 400-m running time | 10m, 20m sprinting ↑, VO2max↑, 400-m running time↑, sprinting↑, Standing long jump↑, |
| Fuentes-García (2021) [36] | Young tennis player | Sex: M, EG = 6, CG = 7, Age:17 ±2 HT: 176.5±4.4, WT: 69.5 ±3.4kg, | Freq.: 2 times/week, time: NR length: 3 mouths Heart rate:80% to 140% Wmax Interval Time:30s Whether it meets the HIIT training intensity and interval time: Yes | High-intensity interval training (EG), intermittent interval training (CG) | Average HR, Lactate, Borg scale, Number of shots, Number of errors | HR↔, Lactate↔, Number of shots↑, Number of errors↑ |
| Fernande et al. (2017) [60] | Recreational Tennis Players | Sex: M, EG = 10, CG = 10, Age: 14±0.1y, WT = 63.8kg, HT: 174.7±4.8cm | Freq.: 2 times/week, time: 15-30min length: 8 weeks Heart rate: NR Interval Time:30s Whether it meets the HIIT training intensity and interval time: Yes | High-intensity interval training (EG), sport-specific drill training (CG) | Laboratory Test, 30–15 Intermittent Fitness Test, Speed Test, Vertical Jumping, | VO2max↑, 5m↔, 10m↔, 20m↔, jumping strength↑ 400m↑ |

EG, experimental group; CG, control group; WT, weight; NR, notreported; HR: Heart rate; HT, height; Freq.,frequency; M, Male;F, Female; YR, year; ↑, significificant within-group improvement from pretest to post-test; ↔, non-significificant within-group change from pretest to post-test.

the papers. Any disagreements that arose during this classification process were thoroughly discussed among all the authors until a unanimous consensus was reached. Utilizing this classification framework, the experimental findings from the ten included studies were systematically compiled and subjected to comprehensive summarization and analysis.

**3.5.1 Effect of HIIT training on speed in <u>racket sports</u> players.** Among the ten studies included in this systematic review, six focused on assessing the impact of HIIT on speed performance. These six studies collectively involved 56 male athletes aged 14 to 18 years [24,36,51,56,59,60]. The speed assessments conducted covered a range of distances, including 5 meters, 10 meters, 20 meters [57], 10 and 400 meters [24,57], and a repeated sprint ability test [57,60]

The studies examined a diverse group of young athletes, including table tennis players [53], badminton players [55,56,58], and tennis players [24,36,54,57,59,60]. Five of these studies reported notable enhancements in straight-line sprint tests. Further improvements were observed in direction change sprint tests [34,60] and sprint ability tests [24,60]. Additionally, significant improvements were noted in the 400-meter run [24]. One study even reported a significant enhancement in batting speed [36].

The results demonstrated statistically significant differences in most performance tests between the pre-intervention and post-intervention periods within the experimental groups. Specifically, in the straight-line sprint tests, effect sizes (d-values) ranged from 0.40 to 1.10 ($p < 0.05$). In the direction change sprint tests, d-values ranged from 0.77 to 0.88 ($p < 0.05$). The repeated sprint ability tests showed a noteworthy reduction in mean sprint time by 3.8% ($p < 0.05$) [57]. In the 400-meter run, the HIIT group's performance significantly increased by 5.2% ($p < 0.05$) [24].

**3.5.2 Effect of HIIT training on strength in <u>racket sports</u> players.** Strength is a pivotal factor in racket sports, as it directly influences skill and tactical performance, subsequently impacting the overall quality of sports performance [24,36,56,59,60]. Athletes in racket sports recognize the significance of enhancing their strength to elevate their performance levels, maintain competitiveness, and refine the precision and quality of their play [24,51,56].

In total, nine studies explored the realm of strength training, involving a combined total of 90 male athletes aged 14 to 18 years, with a mean age of 15.5 ± 2.2 years. Among these studies, some reports indicated a significant enhancement in the ATP-CP system among licensed athletes following HIIT [24]. Additionally, two studies highlighted the substantial impact of HIIT on lower body strength, as demonstrated in jump strength tests and sprint tests [24,60]. Another study reported an overall improvement in athletes' explosive power [54]. However, it's worth noting that one study did not find a significant increase in sprint performance [57].

While numerous studies have evaluated both lower and upper extremity strength, variations often arise due to differences in loading strategies [41,44]. Further research is warranted to delve deeper into the area of strength enhancement in racket sports. In the context of blood lactate concentrations, significant differences were observed between the two time periods (pre-intervention and post-intervention) within the experimental groups during most of the strength tests. These differences were reflected in effect sizes (d-values) ranging from 4.09 ± 0.19 to 3.52 ± 0.76 ($p < 0.05$) for various strength tests, including lower body jumping and sprint tests. However, it's essential to acknowledge that one study did not identify a significant improvement in sprint performance [57].

Given the substantial body of literature examining both lower and upper extremity strength, variations in findings often stem from differences in loading strategies [41,49]. Therefore, additional research efforts are essential for a comprehensive understanding of strength enhancement in this context.

**3.5.3 Effect of HIIT training on endurance in <u>racket sports</u> players.** Endurance requirements in sports vary widely, with physical activities categorized based on their static and dynamic components as well as the involvement of various energy systems. Racket sports, notably, fall into the category of moderate-intensity activities. From this perspective, it's evident that both anaerobic and aerobic energy systems play crucial roles in meeting the energy demands of these sports [10,11,14,61,62].

A total of nine studies in our analysis focused on endurance training, collectively involving 110 male athletes aged 14 to 18 years, with a mean age of 15.5 ± 2.2 years. Notably, Three of the studies showed significant improvements in tennis-specific endurance, with significant reductions in mean HIIT sprint time and improvements in anaerobic capacity [24,36,59].

For badminton players, a similar peak improvement was observed in the 400-meter running time test, accompanied by a substantial 5.2% average increase in vanadium dioxide and a significant 2.4% improvement in performance response. Additionally, there was a notable 6.0% increase in VO2 max levels [55–58].

# 4. Discussion

This systematic review offers an in-depth analysis of the impact of HIIT on key physical attributes—namely strength, speed, and endurance—in racket athletes. Our principal findings underscore significant improvements in athletes' physical performance attributable to HIIT interventions. Of particular note is HIIT's pronounced efficacy in enhancing submaximal endurance performance, as demonstrated by increased running speed and oxygen uptake at various thresholds. Additionally, we observed improvements in repetitive sprint performance and linear sprint running when compared to multiple control regimens.

Strength, speed, and endurance are fundamental components underpinning racket sports success. Given the uniformly positive outcomes seen in these studies, it becomes evident that HIIT serves as an effective intervention for improving the performance of racket athletes. Building on the analytical framework outlined in the Results section, we have carefully scrutinized the variables examined in these studies, thereby elucidating the multifaceted effects of HIIT on the physical capabilities of racket athletes.

## 4.1 Effect of HIIT training on speed in <u>racket sports</u> players

Speed is universally acknowledged as a crucial component of motor skills across various sports disciplines [63,64]. In racket sports, speed's significance is even more pronounced, as it directly affects a player's hitting velocity and, consequently, the match's outcome [36,65]. It is important to note that empirical evidence supporting the efficacy of HIIT in enhancing a tennis player's stroke speed remains relatively limited, with one study standing as an exception [36]. Several studies do suggest, however, that batting speed may be influenced by factors such as motor skill proficiency and muscle contraction velocity [9,36,65].

Additionally, existing research emphasizes the positive impact of HIIT on repetitive sprint performance and linear sprint running [1,24,55,56]. One plausible explanation for these outcomes relates to the nature of HIIT, which places the body in a state conducive to fat loss, muscle retention, and even muscle growth. This, in turn, enlarges the muscle cross-sectional area, improving force transmission and, subsequently, speed [66–68]. Another possible reason centers on HIIT's role in strengthening core muscles [69]. The strategic manipulation of load and intensity enhances the synergy of multiple muscle groups, aiding athletes in maintaining their required center of gravity and coordinating movements, whether for short-term sprints or lateral speed bursts [1,24,55,56].

HIIT consists of high-intensity exercise regimens executed according to a specific plan. Typically, the training pushes the heart rate to exceed 80% of its maximum, with strict interval timing [39]. Generally, two types of HIIT exist: one features high-intensity, low-interval exercises, requiring the heart rate to reach between 100% and 120% of the maximum and lasting for 10–30 seconds, with equivalent rest periods. The other is high-intensity, long-interval training, where the heart rate must reach between 80% and 95% of the maximum and last for 1–3 minutes, with corresponding rest periods [34,44].

A recent study systematically summarized HIIT as characterized by high intensity, heavy load, and interval-limited exercise [18,33]. This form of strength training not only enhances an athlete's speed but also improves the efficiency of complex movements across different sports.

### 4.2 Effect of HIIT training on strength in racket sports players

Research findings on the beneficial effects of HIIT have contributed substantially to our understanding of strength development in racket athletes. Two studies offer compelling evidence that HIIT enhances performance in jump tests among these athletes [24,60]. This assertion is corroborated by literature that includes badminton and tennis players [54,60]. Additionally, several metrics within the jump tests—such as jump height, ground contact time, and peak power—showed statistically significant improvements.

An intriguing parallel can be drawn from a study involving male soccer players, in which the athletes not only improved their strength levels but also achieved greater technical precision through specific HIIT exercises [70]. Collectively, these findings emphasize the positive influence of HIIT interventions on athletes' strength. Furthermore, HIIT's capacity to stimulate athletes' sensory nerves and unlock their latent potential and athleticism—enabling them to excel in competitive environments—is noteworthy.

HIIT primarily functions to augment the body's energy supply capacity within the glycolysis air supply system and the mixed metabolic system of phosphate and glycolysis air supply [13,71,72]. These results contrast with those reported by Wee et al. (2017), which covered badminton players, table tennis players, and tennis players, and suggested that the strength training program under investigation may not have adequately activated the neuromuscular system associated with strength [60,73].

In swimmers, HIIT has been found to positively impact explosive muscle strength [74]. For racket players, the strength of their hitting ability depends not just on the upper arm, shoulder, elbow, and wrist strength during the hitting motion, but also on power transmission from the lower body to the core, as well as the coordinated engagement of various muscle groups in the upper body [8,29,62]. Within this kinetic chain, explosive force plays a vital role [25]. The accelerated running speeds required by HIIT training lead to increased neural drive, resulting in enhanced anaerobic glycolytic activation and the recruitment of additional fast-twitch motor units for brief durations [45]. Consequently, HIIT training programs prove effective in bolstering the muscular power of racket athletes.

### 4.3 Effect of HIIT training on endurance in racket sports players

The influence of HIIT on the endurance of racket athletes can be divided into two components: aerobic and anaerobic endurance [16]. Racket sports are inherently moderate-intensity activities that balance both static and dynamic elements, utilizing both anaerobic and aerobic energy systems [36]. As a result, endurance is a vital attribute for racket athletes, with elite competitors generally displaying superior levels of it [21].

Recent research indicates that even brief bouts of HIIT can significantly improve an individual's maximal oxygen uptake and muscle oxidase activity, thereby enhancing performance

in long-term endurance tasks. From an energy metabolism perspective, anaerobic endurance primarily relies on the phosphagen and glycolysis systems. Notably, HIIT primarily employs the glycolytic energy supply system and a blended metabolic system incorporating both phosphate and glycolysis [35,71,72,75].

One study demonstrated that HIIT significantly improves table tennis players' VO2max and lactic acid uptake in blood [53]. Although HIIT is classified as anaerobic exercise, the after-burn effect induced by intervals allows athletes to engage in continuous, moderate-intensity exercise [18,24]. Moreover, research has shown that HIIT positively impacts the aerobic capacity of racket athletes, particularly by increasing their maximum oxygen uptake and metabolic capacity [36,54,60].

Maximum oxygen uptake, commonly referred to as VO2 max, serves as a key objective measure for assessing both anaerobic and aerobic endurance [70,75,76,77]. Higher oxygen consumption during intense physical activities signifies increased energy production through aerobic metabolism [30,45,68,70]. Numerous studies indicate that HIIT can elevate peak VO2 max among seasoned racket athletes [55–58]. The efficacy of HIIT in boosting maximal oxygen uptake stems from the high-intensity intervals that enable athletes to reach or briefly exceed their anaerobic thresholds, followed by periods of lower aerobic intensity. This pattern overloads the cardiorespiratory system, inducing continuous adaptations in heart and lung functions and ultimately increasing VO2 max.

Moreover, HIIT improves not only aerobic but also anaerobic endurance in racket athletes [22,24,36]. "A study by Ko et al. (2021) found that HIIT elevated the lactate and anaerobic thresholds as percentages of VO2max by four percent compared to athletes who underwent moderate continuous training (MCT) with rackets. This finding aligns with the hypothesis that HIIT can generate higher levels of intensity in the heart, lungs, and muscles, thereby enhancing anaerobic metabolism.

The lactate threshold is a crucial indicator of endurance, marking the point where energy supply shifts from aerobic to anaerobic metabolism. During HIIT sessions, the intensity is so high that oxygen uptake falls short of fulfilling aerobic metabolism demands, causing a surge in anaerobic energy production. This leads to an accumulation of lactic acid, contributing to muscle fatigue [10,11,14,22]. Additional research confirms HIIT's positive effects on anaerobic capacity; it not only increases the body's resilience to lactic acid but also its ability to metabolize and effectively clear or recycle it. This leads to an elevated lactate threshold, meaning that during more intense exercises, there is no significant accumulation of lactic acid, signaling an improvement in anaerobic capacity [24,36,51,55,56,59].

## 5. Limitations

While this review offers valuable insights into the effects of HIIT on speed, strength, and endurance in racket athletes, it is important to acknowledge certain limitations and areas for improvement:

Limited Scope: The existing literature predominantly focuses on HIIT in team-based racket sports like tennis and badminton, with less emphasis on individual sports such as squash and table tennis. Further research addressing the unique needs and benefits of HIIT in these individual sports is warranted.

Comparative Analysis: Although the effectiveness of HIIT has been explored in comparison to traditional training methods, the literature currently lacks a detailed comparison with emerging training techniques. Investigating how HIIT fares against these newer methods could provide a more comprehensive understanding of its impact.

## 6. Conclusion

This systematic review illuminates the growing body of evidence supporting the effectiveness of HIIT in enhancing speed, strength, and endurance in athletes participating in racket sports like tennis, table tennis, and badminton. These findings emphasize HIIT's potential to improve overall athletic performance compared to other training methods. However, it is worth noting that limited research exists on how HIIT specifically influences skill components in racket sports, such as stroke techniques and match strategies. Future research should focus on these areas to offer comprehensive insights that could help athletes improve their skill performance in competitive settings.

## 7. Practical application

In the world of racket sports—be it tennis, badminton, or table tennis—athletes require a diverse skill set that includes speed, strength, endurance, and rapid recovery. These sports demand quick movements, fast reflexes, and sustained performance, making physical conditioning a cornerstone for success. Enter HIIT, a training methodology that has garnered significant attention for its ability to boost athletic performance across various domains. When applied to racket sports, HIIT serves as a potent tool for athletes aiming to elevate their game. Known for its efficiency in expending a substantial amount of energy within a short time frame, HIIT has particular relevance for racket sports, where it can lead to improved physiology, particularly in terms of speed and power. The brief, intense bursts of activity in HIIT align well with the explosive movements required in these sports. By incorporating HIIT into their training regimens, athletes can markedly improve their ability to execute quick, impactful plays on the court or table. Additionally, HIIT has the unique ability to stimulate sensory nerves, contributing to improved proprioception and agility. This enhanced sensory awareness allows athletes to make split-second decisions with greater precision, a vital skill in any racket sport. Metabolically, HIIT aligns well with the energy demands of racket sports, primarily engaging the glycolytic energy system vital for short bursts of high-intensity effort. This form of training mimics the stop-and-start nature of these sports, allowing athletes to optimize their speed, strength, and endurance while managing energy resources efficiently. In summary, HIIT acts as a performance enhancer for racket athletes, fine-tuning their physiological and sensory attributes, and leading to notable improvements in speed, strength, and endurance. Therefore, it is strongly recommended that racket athletes incorporate HIIT into their regular training routines, ideally at a minimum frequency of two sessions per week over four weeks. This structured approach could help athletes unlock their full potential and excel in competitive environments.

## Supporting information

**S1 Checklist. PRISMA 2020 checklist.**
(DOCX)

**S1 Data.**
(DOCX)

## Author Contributions

**Conceptualization:** Yixuan Liu.

**Data curation:** Yixuan Liu.

**Formal analysis:** Borhannudin Bin Abdullah.

**Investigation:** Yixuan Liu.

**Methodology:** Yixuan Liu.

**Project administration:** Borhannudin Bin Abdullah.

**Resources:** Borhannudin Bin Abdullah.

**Supervision:** Hazizi Bin Abu Saad.

**Validation:** Hazizi Bin Abu Saad.

**Visualization:** Hazizi Bin Abu Saad.

**Writing – original draft:** Yixuan Liu.

**Writing – review & editing:** Yixuan Liu.

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
