## [Decision Letter · Decision Letter 0]

16 Oct 2023

PONE-D-23-29988Effects of High-intensity Interval Training on Strength, Speed, and Endurance Performance among Racket Sports Players: A Systematic ReviewPLOS ONE

Dear Dr. Liu,

Thank you for submitting your manuscript to PLOS ONE. After careful consideration, we feel that it has merit but does not fully meet PLOS ONE’s publication criteria as it currently stands. Therefore, we invite you to submit a revised version of the manuscript that addresses the points raised during the review process.

ACADEMIC EDITOR: The reviewers present relevant questions about the study. Please elaborate a point-by-point answer.

We look forward to receiving your revised manuscript.

Kind regards,

Leonardo Vidal Andreato, PhD

Academic Editor

PLOS ONE

Journal Requirements:

Reviewers' comments:

Reviewer's Responses to Questions

**Comments to the Author**

1. Is the manuscript technically sound, and do the data support the conclusions?

Reviewer #1: Yes

Reviewer #2: Yes

2. Has the statistical analysis been performed appropriately and rigorously? 

Reviewer #1: Yes

Reviewer #2: N/A

3. Have the authors made all data underlying the findings in their manuscript fully available?

Reviewer #1: No

Reviewer #2: Yes

4. Is the manuscript presented in an intelligible fashion and written in standard English?

Reviewer #1: No

Reviewer #2: Yes

5. Review Comments to the Author

Reviewer #1: Dear Authors,

The manuscript is well-written and deals with an important topic, and I appreciate the exciting insights your article provides. However, after reviewing the manuscript thoroughly, I have some suggestions for your review. The study has a sound method and content but suffers from many spelling mistakes. Therefore, you need to edit the entire text. Moreover, it is necessary to fully adapt the article to the English language rules.

Page 1: use shortened versions throughout the text (High-Intensity Interval Training) (HIIT)

Page 1: Write which racquet sports they are. “table tennis, tennis, and badminton”, add key words or abstract to the section

Page 2: Reference numbers in the all text are written incorrectly. Check and correct reference numbers throughout the text.

Page 27: use the short form (High-Intensity Interval Training) (HIIT)

References section: check the References (1, 34, 75 control and check the fonts)

Reviewer #2: Review of the article “Effects of high-intensity interval; training on strength, speed and endurance performance among racket sports players: a systematic review”

General Note

In this manuscript, the authors aimed to systematically review the literature on the influence of high-intensity interval training on the strength, speed, and endurance performance of racket sports athletes. After applying the search strategy at four databases and on google scholar and reference lists, the authors included 10 studies in their review. After analyzing the results, the authors emphasizes HIIT as a strategy to improve overall athletic performance comparing to others training methods. The manuscript fits the Plos One scope. Follow bellow some general notes and minor concerns for each section.

General Concerns

Introduction

The introduction session is very extensive. The authors dedicate several pages to approach the problem, justify the investigation and expose the gaps in the specific literature. I agree is utmost important that we have to conduct a “story telling” and “sell the idea”. However, research papers must be objective, precise and direct to the point. It is not the case of the introduction session. The authors used 3 paragraphs to explain HIIT effects on racket sports and, surprisingly, in none of these paragraphs the authors explain what is HIIT. My suggestion to the authors is to collapse the 3 paragraphs in one.

At introduction section, the sentence “Racket athletes often rely on the ATP-CP energy system for short energy bursts,…”. In fact, to perform a specific movement, it is expected that the phosphagens systems were the prominent system to offer energy, however, during all exercise session, the body will migrate from one system to another. As a suggestion, maybe the authors agree to adjust the sentence and be more specific.

Finally, at the end of the introduction section, the authors state that “…research focusing specifically on racket sports remains limited. Notably, there is a lack of systematic reviews examining the physical effects of HIIT on athletes in racket sports.” I believe this sentence is ambiguous since if research in rackets sports remains limited, it is not possible to conduct a robust review with conclusive findings. Maybe the authors consider rewriting the sentences.

Methods

At this section, the authors make a common mistake when trying to review the literature with high-intensity interval exercise. Specific literature on HIIT attested that 11 variables could be manipulated to build a HIIT session. In this sense, is extremely difficult to aggregate HIIT studies to a common analysis since the several possibilities to organize a HIIT protocol. It was expected that at some part of the review, the authors demonstrate and explore the HIIT protocols of the 10 selected. Maybe the author agrees to include in table 3 the adopted protocols. Further, I suggest including why the terminology High-intensity interval training was the chosen one, since it is possible to find studies applying “high intensity interval exercise”; “high intensity intermittent training”; “high intensity intermittent exercise”; “repeated sprint training”; “sprint interval exercise”; among other possibilities.

6. PLOS authors have the option to publish the peer review history of their article (what does this mean?). If published, this will include your full peer review and any attached files.

Reviewer #1: No

Reviewer #2: **Yes: **Marcelo Marques

---

## [Author Response · Author response to Decision Letter 0]

6 Nov 2023

Dear Editors, 

I hope this letter finds you well. I would like to express my gratitude for the thorough review and valuable feedback provided by you and the reviewers on my manuscript titled "Effects of High-intensity Interval Training on Strength, Speed, and Endurance Performance among Racket Sports Players: A Systematic Review." I appreciate the time and effort invested in the evaluation of my work, and I have carefully considered all the comments and suggestions.

I want to address each of the points raised by the academic reviewers in a systematic manner to clarify and, where appropriate, revise the manuscript in line with their recommendations:

Response to Reviewer #1

Dear Professor,

I wish to express my gratitude for your recognition and the thoughtful suggestions you have provided. I would like to offer a concise clarification and an update on the implementation of your suggestions.

Page 1: use shortened versions throughout the text (High-Intensity Interval Training) (HIIT)

The abbreviation has been changed

Page 1: Write which racquet sports they are. “table tennis, tennis, and badminton”, add key words or abstract to the section

Already added to keywords

Page 2: Reference numbers in the all text are written incorrectly. Check and correct reference numbers throughout the text.

The number has been rechecked

Page 27: use the short form (High-Intensity Interval Training) (HIIT)

The abbreviation has been changed

References section: check the References (1, 34, 75 control and check the fonts)

Reference has been modified

Response to Reviewer #2

Dear Professsor, 

I appreciate your thorough review; it is truly an honor. And i have implemented revisions in the manuscript in accordance with your suggestions, and I will provide clarifications below:

Introduction

The introduction session is very extensive. The authors dedicate several pages to approach the problem, justify the investigation and expose the gaps in the specific literature. I agree is utmost important that we have to conduct a “story telling” and “sell the idea”. However, research papers must be objective, precise and direct to the point. It is not the case of the introduction session. The authors used 3 paragraphs to explain HIIT effects on racket sports and, surprisingly, in none of these paragraphs the authors explain what is HIIT. My suggestion to the authors is to collapse the 3 paragraphs in one.

At introduction section, the sentence “Racket athletes often rely on the ATP-CP energy system for short energy bursts,…”. In fact, to perform a specific movement, it is expected that the phosphagens systems were the prominent system to offer energy, however, during all exercise session, the body will migrate from one system to another. As a suggestion, maybe the authors agree to adjust the sentence and be more specific.

Finally, at the end of the introduction section, the authors state that “…research focusing specifically on racket sports remains limited. Notably, there is a lack of systematic reviews examining the physical effects of HIIT on athletes in racket sports.” I believe this sentence is ambiguous since if research in rackets sports remains limited, it is not possible to conduct a robust review with conclusive findings. Maybe the authors consider rewriting the sentences.

I have incorporated an introduction to HIIT within the introductory section and have combined the three relevant paragraphs into a single unified paragraph, as per your guidance. Additionally, I have provided a comprehensive explanation of the connection between racket players and the ATP-CP energy system. Furthermore, I have addressed the sentence modification you highlighted at the end of the text.

Methods

At this section, the authors make a common mistake when trying to review the literature with high-intensity interval exercise. Specific literature on HIIT attested that 11 variables could be manipulated to build a HIIT session. In this sense, is extremely difficult to aggregate HIIT studies to a common analysis since the several possibilities to organize a HIIT protocol. It was expected that at some part of the review, the authors demonstrate and explore the HIIT protocols of the 10 selected. Maybe the author agrees to include in table 3 the adopted protocols. Further, I suggest including why the terminology High-intensity interval training was the chosen one, since it is possible to find studies applying “high intensity interval exercise”; “high intensity intermittent training”; “high intensity intermittent exercise”; “repeated sprint training”; “sprint interval exercise”; among other possibilities.

I have incorporated Table 3, as well as the two crucial components of the intervention, namely heart rate and interval time, as per your guidance. These two variables play a pivotal role in defining the exercise intensity and can serve as indicators of whether the experiment aligns with the criteria for HIIT. 

In the introduction, I have also emphasized the rationale for adopting the term 'HIIT,' underlining why it is the recommended descriptor. The following is my definition and explanation of this issue.

High-Intensity Interval Training (HIIT), "high intensity interval exercise," "high intensity intermittent training," and "high intensity intermittent exercise" are terms that are often used interchangeably. They all describe the same fundamental concept: alternating high-intensity exercise with rest intervals or lower-intensity exercise.

On the other hand, Repeated Sprint Training is a specific type of training primarily used in sports involving sprinting or short bursts of intensity. It typically involves repeated, brief, maximal-effort sprints with short recovery periods between each sprint. While it shares similarities with HIIT, Repeated Sprint Training is often tailored for athletic events and is particularly focused on enhancing sprinting capabilities.

Similarly, Sprint Interval Exercise involves short bursts of maximal effort sprints followed by rest or low-intensity intervals. Like Repeated Sprint Training, it is commonly employed in sport conditioning programs.

The term HIIT has achieved widespread recognition as the preferred and widely recognized phrase for this specific training method. This is due to its clarity and consistency. When individuals hear "HIIT," they immediately grasp the training regimen characterized by short, intense bursts of exercise alternated with periods of rest or low-intensity intervals. Over time, it has garnered immense popularity, making it a powerful and easily identifiable concept within the fitness industry.

Secondly, HIIT offers simplicity. It is a succinct and clear acronym that effectively summarizes the core elements of this training method. Its ease of recall and application make it highly accessible.

Furthermore, HIIT has garnered substantial recognition within academic and research disciplines over the long term. It has become a cornerstone of scientific investigations and studies, solidifying its position as the preferred term for describing this form of training. Researchers and academics frequently rely on standardized terminology to ensure precision and clarity in their work.

In summary, these terms all describe training methods that involve alternating periods of high-intensity exercise with rest or low-intensity intervals. The choice of terminology may vary depending on the context, the specific goals of the training program, or the sport or fitness discipline in question. However, the fundamental principles of HIIT remain consistent across these terms.

I believe that these revisions have significantly improved the quality and clarity of the manuscript. 

If further revisions are necessary or if you have any additional questions or concerns, please do not hesitate to contact me. I am committed to ensuring that the manuscript meets the journal's requirements and maintains its academic rigor.

Thank you once again for your time and expertise in this review process. I look forward to hearing from you and the reviewers regarding the outcome of this submission.

Sincerely,

Yixuan Liu

---

## [Decision Letter · Decision Letter 1]

20 Nov 2023

Effects of High-intensity Interval Training on Strength, Speed, and Endurance Performance among  Racket Sports  Players: A Systematic Review

PONE-D-23-29988R1

Dear Dr. Liu,

We’re pleased to inform you that your manuscript has been judged scientifically suitable for publication and will be formally accepted for publication once it meets all outstanding technical requirements.

Kind regards,

Leonardo Vidal Andreato, PhD

Academic Editor

PLOS ONE

Additional Editor Comments (optional):

Reviewers' comments:

Reviewer's Responses to Questions

**Comments to the Author**

1. If the authors have adequately addressed your comments raised in a previous round of review and you feel that this manuscript is now acceptable for publication, you may indicate that here to bypass the “Comments to the Author” section, enter your conflict of interest statement in the “Confidential to Editor” section, and submit your "Accept" recommendation.

Reviewer #2: All comments have been addressed

2. Is the manuscript technically sound, and do the data support the conclusions?

Reviewer #2: Yes

3. Has the statistical analysis been performed appropriately and rigorously? 

Reviewer #2: N/A

4. Have the authors made all data underlying the findings in their manuscript fully available?

Reviewer #2: Yes

5. Is the manuscript presented in an intelligible fashion and written in standard English?

Reviewer #2: Yes

6. Review Comments to the Author

Reviewer #2: Congrats for the manuscript. It was a pleasure to review this revised version. I hope I contributed to this manuscript.

7. PLOS authors have the option to publish the peer review history of their article (what does this mean?). If published, this will include your full peer review and any attached files.

Reviewer #2: **Yes: **Marcelo Marques

---

## [Editor Report · Acceptance letter]

4 Dec 2023

PONE-D-23-29988R1 

Effects of High-intensity Interval Training on Strength, Speed, and Endurance Performance among Racket Sports Players: A Systematic Review 

Dear Dr. Liu:

I'm pleased to inform you that your manuscript has been deemed suitable for publication in PLOS ONE. Congratulations! Your manuscript is now with our production department. 

Kind regards, 

on behalf of

Dr. Leonardo Vidal Andreato 

Academic Editor

PLOS ONE